# Host-Adapted Gene Families Involved in Murine Cytomegalovirus Immune Evasion

**DOI:** 10.3390/v14010128

**Published:** 2022-01-11

**Authors:** Sara Becker, Annette Fink, Jürgen Podlech, Matthias J. Reddehase, Niels A. Lemmermann

**Affiliations:** Institute for Virology and Research Center for Immunotherapy (FZI), University Medical Center of the Johannes Gutenberg-University Mainz, 55131 Mainz, Germany; sarbecke@uni-mainz.de (S.B.); annette.fink82@t-online.de (A.F.); podlech@uni-mainz.de (J.P.); lemmermann@uni-mainz.de (N.A.L.)

**Keywords:** adoptive cell transfer, antigen presentation, antiviral protection, CD8 T cells, co-evolution, cytomegalovirus, gene family, immune evasion, latent infection/latency, memory inflation

## Abstract

Cytomegaloviruses (CMVs) are host species-specific and have adapted to their respective mammalian hosts during co-evolution. Host-adaptation is reflected by “private genes” that have specialized in mediating virus-host interplay and have no sequence homologs in other CMV species, although biological convergence has led to analogous protein functions. They are mostly organized in gene families evolved by gene duplications and subsequent mutations. The host immune response to infection, both the innate and the adaptive immune response, is a driver of viral evolution, resulting in the acquisition of viral immune evasion proteins encoded by private gene families. As the analysis of the medically relevant human cytomegalovirus by clinical investigation in the infected human host cannot make use of designed virus and host mutagenesis, the mouse model based on murine cytomegalovirus (mCMV) has become a versatile animal model to study basic principles of in vivo virus-host interplay. Focusing on the immune evasion of the adaptive immune response by CD8^+^ T cells, we review here what is known about proteins of two private gene families of mCMV, the *m02* and the *m145* families, specifically the role of *m04*, *m06*, and *m152* in viral antigen presentation during acute and latent infection.

## 1. Introduction

Human cytomegalovirus (hCMV), the prototype member of the β-subfamily of the herpes virus family [1], is of medical relevance as it can cause lethal disease resulting from cytopathogenic organ infection in an immunologically immature or immunocompromised human host (for a synopsis of hCMV disease manifestations, see [2,3]). One area of concern and significant public health impact are fetuses infected upon diaplacental transmission of the virus in expectant mothers primarily infected, superinfected, or reactivating latent virus during pregnancy. This congenital infection results in a high risk of developing birth defects with long-term sequelae in survivors, such as sensorineural hearing loss and mental retardation (for reviews, see [4,5]). In addition, in transplantation centers worldwide, primary or reactivated hCMV is a feared complication in transplantation patients. These include recipients of solid organ transplantation (SOT) who receive immunosuppressive prophylaxis to prevent allograft rejection (for reviews, see [6,7]) as well as recipients of hematopoietic cell transplantation (HCT) who are immunocompromised by hematoablative therapy of an aggressive hematopoietic malignancy prior to HCT and, in particular in the case of unrelated or family donors, by immunosuppressive prophylaxis or therapy of graft-versus-host disease (GvHD) after HCT [8,9,10]. Graft loss in SOT and an often lethal interstitial pneumonia in HCT are major disease manifestations of hCMV infection. In fact, follow-up monitoring for early detection of hCMV reactivation and controlling the success of antiviral therapy in HCT recipients is a major assigned task of virological laboratory diagnostics at university centers that operate an HCT unit.

One pathogenetic factor of all CMVs is the evolutionary acquisition of genes that subvert recognition of infected cells by the hosts’ intrinsic pathogen sensing mechanisms as well as the innate and adaptive immune response (for reviews, see [11,12,13,14,15]). As a consequence of adaptation to the respective mammalian host during the estimated 350 million years of co-evolution [1], CMV species differ from each other in so-called “private genes”. These are defined by the absence of sequence homologs in phylogenetically distant CMV species, except for some overlap between hCMV and the phylogenetically less distant non-human primate CMVs [1]. It appears that duplications and mutations starting from an ancestor gene, supposedly captured from the host early in co-evolution, were the viral evolutionary response to the development of polymorphism of cellular genes involved in host immunity, such as genes encoding antigen-presenting HLA/MHC molecules. This may explain in part why private genes are often organized in gene families, whose corresponding proteins share motifs for cargo sorting and/or are structurally related to host proteins for cellular cargo binding. Gene families involved in immune evasion have been comprehensively reviewed by Berry and colleagues [15]. Specifically, immune evasion proteins of hCMV cluster in gene families (i) *US2-US11*, sharing an immunoglobulin domain and interfering with MHC class I (MHC-I) cell surface trafficking, (ii) the *UL18* family of MHC-I homologs, (iii) the less conserved *RL11* family of mostly undefined precise function, and (iv) the *US12* family thought to be involved in the regulation of NK cell ligands, adhesion molecules, and cytokine receptors (for greater detail, see [15,16]).

An in vivo study of hCMV immune evasion by clinical investigation in the human host is limited, because designed mutagenesis of both the virus and the host is prohibited for ethical reasons or is unfeasible. Despite significant genetic differences in both virus and host to always bear in mind, this is the justification for non-primate, usually rodent, CMV models. Ideally, studies in animal models should address questions that cannot be answered by clinical investigation in the human host, such as cooperative or antagonistic effects of immune evasion proteins tested by combinatorial viral gene deletion. As we have reviewed previously, results from the mouse model of infection with murine CMV (mCMV) have revealed basic principles of CMV pathogenesis, immune control, and immune evasion that also apply to human infection, although not in molecular details but by convergence in many aspects of virus-host interaction [17]. It should be recalled that immune evasion of CMVs was first described in the mouse model [18,19].

Here we review what is currently known about members of the mCMV gene families *m02* and *m145* that localize to opposite ends of the viral genome (Figure 1). Focus is given to the role of proteins encoded by genes *m04* and *m06* of the *m02* gene family and *m152* of the *m145* gene family in positive and negative regulation of MHC-I trafficking and cell surface presentation of peptide-loaded MHC-I (pMHC-I) molecules for recognition by antivirally protective CD8^+^ T cells during acute and latent infection.

## 2. The *m02* Gene Family

Known key features of *m02* gene family members are compiled in Table 1.

The m02 family members usually represent type-I integral membrane glycoproteins that primarily localize in the endoplasmic reticulum (ER), most of which carry a cargo sorting motif in their C-terminal, cytosolic tail. A canonical YXXΦ motif (proteins m02, m04, m05, m07–m10, and m16) can link the protein to heterotetrameric cellular adapter proteins (AP) AP-2 and/or AP-4, whereas a di-leucine motif E(X)XXXLL (protein m06 only) can link the protein to AP-1 and/or AP-3 [39,40,41]. Although the di-leucine motif in m06 differs from the canonical motif by an additional spacer residue, it has proven functional, based on loss of function after replacing LL with AA [33,36]. If the noncanonical tyrosine-based motif in m12 is functional awaits mutational analysis. Notably, the motifs in m04 and m06 are highly conserved among mCMV isolates/strains sequenced so far, which suggests an important function of these motifs in virus-host co-evolution (Figure 2).

Structural analysis of m04 as an m02 family paradigm revealed a shared β-sandwich immunoglobulin variable (Ig-V)-like fold in the luminal (N-terminal) domains of m04 and m06 to where pMHC-I cargo binds [24,47]. Based on the structural analysis, Berry and colleagues suggested an m04-like fold for all family members except m12 and m13 [24]. For m06, NMR-based studies by Sgourakis and colleagues [48] revealed tight binding to pMHC-I L^d^ at a discrete site located underneath the peptide-binding platform that partially overlaps with the β2-microglobulin interface on the MHC-I heavy chain.

Thus, by binding to pMHC-I in the ER and motif-dependent linking of the complex to the respective cellular APs for cargo sorting, m04 and m06 determine the intracellular trafficking and fate of pMHC-I. For the remaining family members (m02, m05, m07–m10, and m16) that also carry YXXΦ sorting motifs, the cargos have not yet been defined. Theoretically, it could be cargos not involved in innate or adaptive immune responses, though MHC-I molecules of *H-2* haplotypes other than those binding to m04 remain candidates. One often tends to forget that virus evolution is driven by adaptation to host evolution and thus must cover MHC polymorphism. Traditionally, studies in the mouse model focus on haplotypes *H-2^b^* and *H-2^d^*, and less often *H-2^k^* is studied, whereas other *H-2* haplotypes are rarely considered. As discussed by Corbett and colleagues [23], high sequence variability of m04 among mCMV isolates, except for the conserved endocytic motif (Figure 2), might reflect an adaptation to MHC-I polymorphism. The structural analysis of m04 performed by Berry and colleagues [24] included variants m04^Smith^, m04^G4^, and m04^W8211^, and revealed their binding to the MHC-I molecules D^d^, L^d^, and D^k^. As already shown experimentally in earlier reports, m04^Smith^ binds to D^d^, L^d^, and K^b^, but not to K^d^ [25,26,27]. It is thus open to question as to which of the many untested MHC-I molecules of other inbred mouse *H-2* haplotypes or wild-derived mice m04 might bind or fail to bind (for a list of *H-2* haplotypes and MHC-I alleles, see http://www.imgt.org/IMGTrepertoireMH/ [49], accessed on 8 January 2022). We, therefore, speculate that other *m02* family members that carry an AP-2 binding YXXΦ motif take on an m04-equivalent role for MHC-I molecules of *H-2* haplotypes not targeted by m04.

The divergent pathways of intracellular sorting determined by the AP binding motifs already suggest that proteins m04 and m06 act antagonistically and compete for pMHC-I cargo. In fact, pMHC-I captured by m04 traffics to the cell surface [25], where the complex binds to the endocytic adapter AP-2 that internalizes it by mediating clathrin-assisted endocytosis [29,50,51,52], whereas binding by m06 ends up in lysosomal disposal [33,34,35]. Inactivating the m06 di-leucine motif by replacing LL with AA was found to block the transition of m06-MHC-I complexes from early endosomes to late endosomes, and thus prevent lysosomal degradation [36].

The period during which m04-pMHC-I complexes are displayed at the cell surface is apparently long enough to silence NK cells by binding to inhibitory Ly49 family receptors Ly49A, Ly49C, and Ly49G2 [25,26,27,28,29,30]. Inhibition of AP-2 mediated endocytosis by a motif mutation Y248A, replacing the crucial Tyr with Ala, was found to intensify NK cell silencing, in accordance with an extended cell surface half-life of m04-pMHC-I complexes [29].

So, m04 can be viewed as being an innate immunity NK cell immunoevasin. However, an MHC-I allele-specific exception exists: cell surface display of m04-D^k^ activates NK cells by binding to the activatory Ly49 family receptors Ly49P1, Ly49L, and Ly49D2 [53,54]. This complexity in NK cell silencing or activation again highlights the importance of MHC-I polymorphism in virus-host co-evolution.

As already shown by Lu and colleagues [27], m04 can bind to empty MHC-I K^b^ in cell lines deficient in peptide-transport from the cytosol into the ER by deletion of the TAP transporter, but the presence of a K^b^-presented viral peptide stabilized the complex. So, m04-pMHC-I complexes travel to the cell surface and have the potential to present viral peptides to CD8^+^ T cells for mediating antiviral control. Although the structural analysis by Berry and colleagues [24] has not precisely mapped the m04 binding site to the MHC class-I α-chain, the data suggest that the peptide-binding platform is not sterically masked, so that the TCR of CD8^+^ T cells should be able to recognize presented peptide. Nonetheless, it was previously speculated that m04 might act as an immunoevasin by preventing the m04-pMHC complex from being recognized by the TCR [26]. As a further potential sterical hindrance, viral protein MATp1 was found to be essential for m04-mediated MHC-I cell surface trafficking [30].

Using combinatorial deletion of the discussed mCMV immunoevasins m04, m06, and m152 (see below) in the mCMV genome revealed an anti-immunoevasive role of m04 in recognition of infected cells by CD8^+^ T cells [31,55], which was confirmed more recently after retrospective verification of the authenticity of the viral mutants involved [32]. This finding prompted us to suggest the acronym vRAP (viral regulator of antigen presentation) to account for both negative and positive roles of viral proteins in pMHC-I cell surface trafficking and presentation to CD8^+^ T cells.

## 3. The *m145* Gene Family

Known key features of *m145* gene family members are compiled in Table 2.

Members of the m145 family usually represent MHC-I-like virally encoded (MHC-Iv) integral type-I membrane glycoproteins that interact with classical MHC-I (α chain) or with cellular MHC-I-like proteins. Specifically, the crystal structure of the complex between m152/gp40 of mCMV and the cellular RAE1γ ligand of the activatory NK cell receptor NKG2D revealed a paradigm for MHC/MHC interaction in immune evasion [57]. Some m145 family members mediate NK cell evasion by downmodulation of NKG2D ligands. Examples are m145, m152, and m155 that bind to and thus downmodulate MULT-1, RAE-1 family members, and H60, respectively (Table 2). Particular attention in high-ranking journals was paid to m157, which activates the Ly49H^+^ subset of NK cells by direct m157-Ly49H interaction, which results in just the opposite of viral immune evasion. However, the evolutionary advantage for the virus is unclear, and in fact, the phenomenon is limited to the C57BL/6 laboratory mouse strain, in which Ly49H^+^ cells are present. Most important to this discussion is the work by McWhorter and colleagues [82], who have shown in C57BL/6 mice that under conditions of within-host competition between co-infecting strains of mCMV, those which expressed an m157 gene product capable of ligating Ly49H were outcompeted by strains expressing variant m157 proteins unable to ligate Ly49H. Thus, in virus evolution, m157 is maintained in its original sequence only in wild-derived mice not expressing Ly49H, and its actual biological function in virus-host co-evolution is still open to question.

The mCMV m152/gp40 glycoprotein was the first immune evasion protein identified for a CMV by the group of U.H. Koszinowski [18,19,58]. Of particular interest is the finding that it simultaneously mediates evasion of CD8^+^ T cells and NK cells by downmodulating cell surface expression of classical pMHC-I and of RAE-1 family ligands of the activatory NK cell receptor NKG2D, respectively [61].

Regarding the mechanism, m152 binds to luminal domains of its interaction partners in the ER and traps the complexes in a cis-Golgi/ER-Golgi intermediate compartment (ERGIC) [83,84,85,86]. It exists in differentially glycosylated isoforms, but the unglycosylated p36 proved to be sufficient for its retention function [84]. The crystal structure of the m152-RAE1γ complex indicated direct binding of the α1/α2 helices of RAE1 to the α1/α2 and α3 domains of m152 [57], and also predicted direct binding of m152 to the α1/α2 domains of the MHC-I α chain. Whereas earlier work proposed a transient contact [83], experimental evidence for true binding was presented more recently [85]. The linker between the luminal and the transmembrane domain of m152 proved important for MHC-I retention. Replacing the authentic 43-amino-acid linker [57] with a (GGGGS)_9_ tandem sequence led to rapid degradation of m152 and loss of its MHC-I retention function [85], resulting in normal levels of pMHC-I cell surface expression. TMED10, a transmembrane protein involved in COPI- and COPII-mediated vesicular export, binds to the linker sequence of m152 and thereby anchors m152 to the ERGIC, thus retaining bound pMHC-I in the ERGIC [85,86].

Recent work has shown an additional function of m152 by its targeting of STING, which plays a central role in intrinsic immunity. After activation by the DNA-sensing molecule cGAS, STING dimerizes and travels from the ER to the Golgi, where it is responsible for the activation of IRF3 and NFκB, resulting in the induction of type-I IFNs and proinflammatory cytokines. Translocation of STING from the ER to the Golgi is slowed down by m152, ultimately leading to a decreased translocation of IRF3 to the nucleus and, consequently, to reduced expression of antiviral type-I IFNs [65].

## 4. The Immune Evasion Enigma in Cytomegalovirus Memory Inflation

The function of viral immune evasion molecules has been established in models of acute, productive infection. Hallmarks of CMV infections are the establishment of latent infection, referred to as “latency”, in certain cell types after clearance of the productive infection (for a review, see [87]) as well as “memory inflation” (MI) that is associated with latency (for reviews, see [88,89]). MI is defined as the expansion of the pool of virus epitope-specific inflationary T effector-memory CD8^+^ T cells (iTEM) that are characterized by the cell surface phenotype KLRG1^+^CD62L^-^ [90]. Maintenance of KLRG1^+^ expression by iTEM depends on repetitive antigen presentation during latent infection [91,92,93,94]. Antigen-driven MI during latency raised the question of why immune evasion molecules m06 and m152 fail to prevent the presentation of pMHC-I to inflationary iTEM.

The solution to this enigma was provided only recently through analysis of viral transcription patterns in latently infected lungs (Figure 3) [95]. In general thinking, it was implicitly assumed that MI is driven by virus reactivation from latency [96], even though completion of the viral replicative cycle by virion assembly and release was shown to be not essential for MI to occur [97]. According to this “reactivation hypothesis”, antigenic viral peptides are derived from proteins synthesized in a cell during a reactivated viral replication cycle that is characterized by temporally coordinated gene expression progressing from immediate–early (IE) to early (E) and late (L) gene expression. Yet, if this applies, immune evasion proteins, which belong to the kinetic class of E proteins, are inevitably expressed in every cell in which antigenic proteins are processed and should thus inhibit pMHC-I cell surface presentation and iTEM restimulation. As an alternative, the “stochastic expression hypothesis” predicted a random and transient desilencing of individual genes in still latent viral genomes, so that antigen-encoding genes and immune evasion genes are rarely expressed coincidentally in the same cell. This allows antigen presentation and restimulation of iTEM by cells in which antigen-encoding genes are desilenced while immune evasion genes remain silenced (Figure 3A). Analysis of viral transcription in lung tissue pieces of latently infected HCT recipients in the mouse model (Figure 3B) indeed revealed random patterns of gene expression in support of the “stochastic expression hypothesis” (Figure 3C) [95]. In perspective, future experiments should address the question of whether stochastic expression of immunoevasins m06 or m152 in a latently infected cell activates or silences NK cells by the downmodulation of cell surface MHC-I loaded with cellular self-peptides (“missing self” activation) or of ligand RAE-1 of the activatory NK cell receptor NKG2D, respectively.

## 5. Concerted Action of vRAPs in Acute Infection

Unlike in latency, when stochastic gene expression can lead to isolated expression of individual immune evasion genes, presentation of antigenic peptides as pMHC-I complexes at the cell surface is regulated by a concerted action during acute infection. To unravel the interplay between the positive vRAP m04 and the negative vRAPs m06 and m152, recent work pursued the strategy of stepwise combination [32] (Figure 4). Antigen presentation was assessed in vivo in immunocompromised C57BL/6 (*H-2^b^*) mice through control of liver infection after adoptive transfer of CD8^+^ cytolytic T lymphocytes (CTL) specific for the antigenic viral peptide M45-D^b^ {985-*HGIRNASFI*-993} presented by MHC-I D^b^ [98] (M45-D^b^ CTL).

This peptide was chosen for the analysis as its presentation is utmost sensitive to regulation. It represents the prototype of a “non-protective epitope”, as originally defined by the failure of M45-D^b^ CTL to control infection by wild-type (WT) virus expressing all three vRAPs, despite an exquisitely high functional avidity of the CTL used [60,98,99]. The explanation for this observation is the very low efficacy of antigen processing generating this peptide, leading to a yield of only ~10 molecules per WT-virus infected cell, compared to ~6000 molecules of the protective viral peptide M45-D^d^ {507-*VGPALGRGL*-515} processed from the same protein and presented by MHC-I D^d^ [99]. Notably, the deletion of just vRAP m152 in virus mCMV-Δm152 restored M45-D^b^ presentation in infected cells as well as in vivo antiviral protection by M45-D^b^ CTL upon adoptive transfer [60,99]. Likewise, IFNγ-enhanced processing in WT-virus infected cells generated ~1000 molecules of the M45-D^b^ peptide, resulting in cell surface presentation sufficient for recognition by high-avidity M45-D^b^ CTL. Accordingly, adoptive transfer of M45-D^b^ CTL protected against in vivo WT virus infection in transgenic B6-SAP-IFNγ mice constitutively producing high serum levels of IFNγ [100].

In the study of the concerted action of the three vRAPs in immunocompromised adoptive transfer recipients (Figure 4), expression of just m152 by mCMV-Δm04m06 completely prevented protection by M45-D^b^ CTL, whereas the addition of m04 restored protection in recipients infected with mCMV-Δm06. This positive regulatory function of m04 was almost completely antagonized by the addition of vRAP m06 in recipients infected with the WT virus (Figure 4A). Corresponding 2-color immunohistological images of liver tissue sections show a high degree of infection with only a few tissue-infiltrating CD8^+^ T cells, that are M45-D^b^ CTL, after infection with mCMV-Δm04m06, whereas additional expression of m04 by mCMV-Δm06 recruits M45-D^b^ CTL to infected cells, thereby forming nodular inflammatory foci (NIF), which are known histological correlates for viral epitope-specific protective activity [101]. Finally, additional expression of vRAP m06 by WT-virus prevented protective NIF formation (Figure 4B).

## 6. Synopsis of vRAP Interplay in Regulating Antigen Presentation

It is the main function of vRAPs to direct recently peptide-loaded pMHC-I complexes from the ER into sorting pathways [102]. Accordingly, they bind to their pMHC-I cargo in the ER and determine its intracellular trafficking. When expressed in the absence of vRAPs m04 and m06 after infection with mCMV-Δm04m06, vRAP m152 has the chance to bind to essentially all recently-loaded pMHC-I complexes and traps them in the ERGIC, so that cell surface presentation is largely prevented (Figure 5, left). When vRAP m04 comes additionally in case of infection with mCMV-Δm06, it successfully competes with m152 for pMHC-I molecules, presumably by binding with higher affinity, and rescues them from getting trapped in the ERGIC, which results in cell surface presentation of m04-pMHC-I complexes. This rescue is incomplete, as m04 is expressed later than m152 in the viral gene expression program. By binding to cellular adapter protein AP-2 through the endocytic YXXΦ motif in the cytosolic tail of m04, m04-pMHC-I complexes are removed from the cell surface by clathrin-assisted endocytosis, which reduces their cell surface presentation half-life (Figure 5, center). Finally, when m06 comes in addition in the case of infection with WT virus, it competes with m04 for pMHC-I binding and directs part of them into the endosomal-lysosomal pathway for degradation by linking the complexes to cellular adapter proteins AP-1A and AP-3A through the di-leucine motif in its cytosolic tail (Figure 5, right).

Immune evasion conveys an evolutionary advantage to virus variants upon host co-infection with multiple variants competing for virus spread in host organs and host-to-host transmission. The finding that m152 is sufficient for preventing the presentation of antigenic peptides to CD8^+^ T cells raises the question of why successful virus variants have additionally acquired the complicated antagonism between m04 and m06. One idea for discussion is that m04 was acquired to counter-act “missing self” activation of NK cells by restoring “self” for NK cell silencing. This function in NK evasion, however, comes at the expense of evading CD8^+^ T cells. To avoid this, m06 was acquired to limit cell surface expression of m04-pMHC-I complexes, thereby balancing and optimizing both NK cell and CD8^+^ T cell evasion.

While the acquisition of vRAPs is to the advantage of the virus and has thus driven viral evolution, immune evasion can result in severe medical problems. As we have shown recently, vRAP-mediated reduction in viral antigen presentation is responsible for lethal viral pathogenesis from unrestricted virus spread in mouse models of major or minor histocompatibility antigen mismatched HCT ([104,105], reviewed in [106]).

## 7. Conclusions and Outlook

More and more, the picture emerges that the interplay between the different viral immune evasion proteins has evolved to balance innate immunity by NK cells and adaptive immunity by T cells. Currently, however, our view is narrowed by low MHC polymorphism coverage of mouse models preferentially used in experimental studies. In Table 1 and Table 2, we have compiled what is known about the members of the *m02* and *m145* gene families, respectively, mostly based on studies of MHC haplotypes *H-2^b^*, *H-2^d^*, and *H-2^k^*. The gaps in the lists are not of less importance, however. The as yet unidentified roles of many of the gene family members may further refine the picture and complete it in terms of viral adaptation to MHC polymorphism.

## Figures and Tables

**Figure 1 viruses-14-00128-f001:**
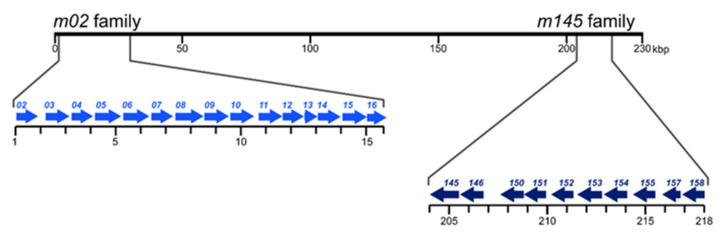
Map positions of mCMV gene families *m02* and *m145*. Arrows indicate the direction of transcription on the coding DNA strand [20,21]. In mCMV gene nomenclature, lower case “*m*” indicates an absence of homology to hCMV genes.

**Figure 2 viruses-14-00128-f002:**
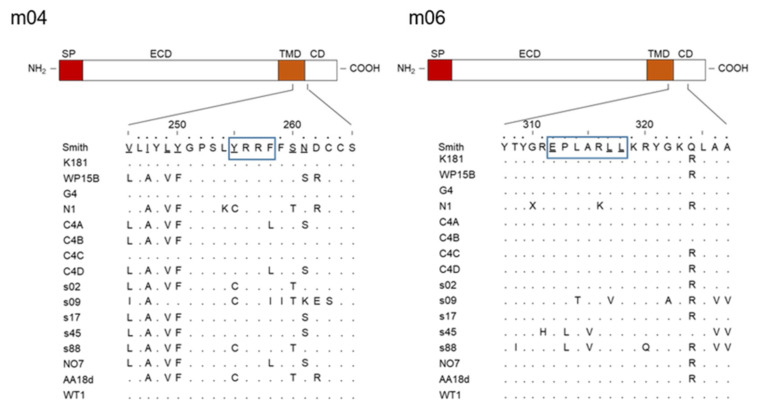
High evolutionary conservation of *m04* and *m06* cargo sorting motifs in strains of mCMV. (Smith NC_004065 [20], K181 AM886412 [42], WP15B EU579860 [43], G4 EU579859 [43], N1 HE610454 [44], C4A EU579861 [43], C4B HE610452 [44], C4C HE610453 [44], C4D HE610456 [44], s02 MH118557 [45], s09 MH118555 [45], s17 MH118558 [45], s45 MH118556 [45], s88 MG957497 [45], NO7 HE610455 [44], AA18d HE610451 [44], WT1 GU305914 [46]).

**Figure 3 viruses-14-00128-f003:**
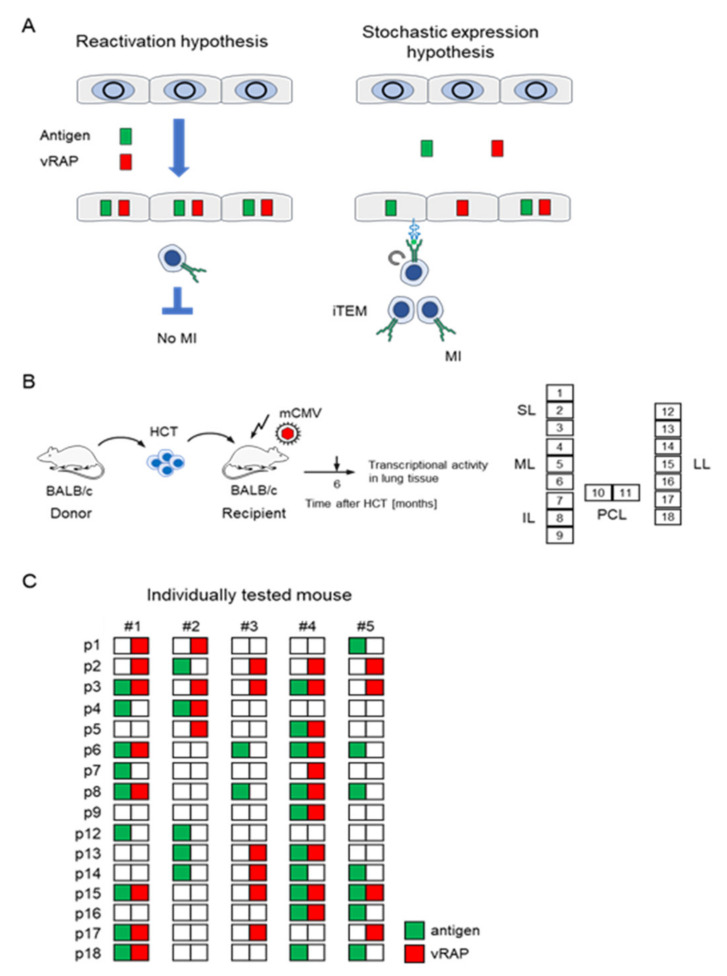
(**A**) Alternative models of viral gene expression during latency. (Circle) latent viral genome in the nucleus of a latently infected cell. (Antigen) IE1 and/or m164 peptide; (vRAP) m06 and/or m152; (MI) memory inflation; (iTEM) inflationary T effector-memory CD8^+^ T cell. (**B**) BALB/c (*H-2^d^*) HCT model for establishing mCMV latency in the lungs. For the analysis of transcripts by RT-PCR, lungs are cut into 18 pieces, p1-p18; (SL, ML, IL) superior, middle, and inferior lobe of the right lung; (PCL) post-caval lobe; (LL) left lung. (**C**) Experimentally observed random transcription patterns in 5 mice (#1–#5) tested individually. Reproduced from reference [95] in a new arrangement.

**Figure 4 viruses-14-00128-f004:**
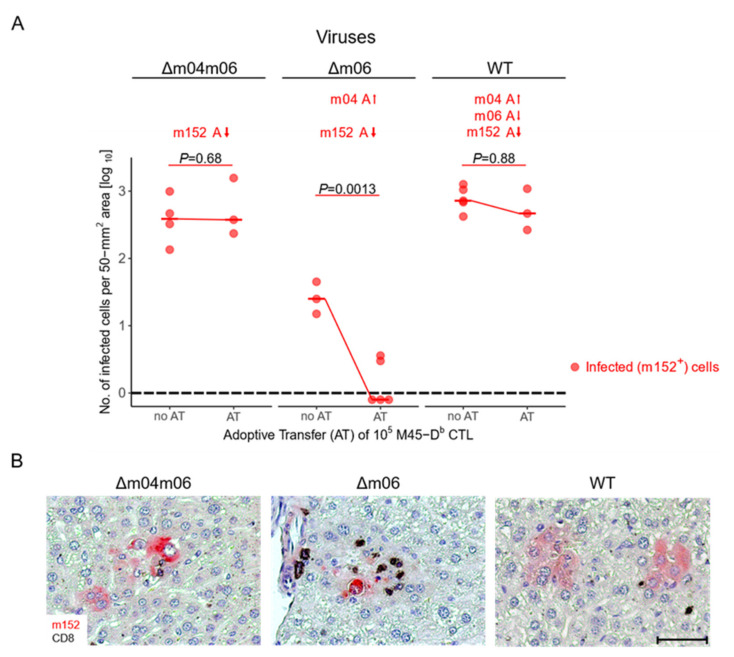
(**A**) Adoptive transfer of M45-D^b^ CTL into immunocompromised C57BL/6 mice infected with viruses mCMV-Δm04m06, selectively expressing vRAP m152, mCMV-Δm06 expressing vRAPs m152 and m04, as well as wild-type (WT) mCMV expressing vRAPs m152, m04, and m06. Data represent infected m152^+^ cells present in 50-mm^2^ areas of liver tissue sections. The red dot symbol represents mice tested individually. Median values are marked and connected for comparison. (A-arrow up or down), positive or negative effect on antigen presentation, respectively; (AT) adoptive transfer. *P* values for comparing groups with or without AT were calculated by a Student’s *t*-test with Welch’s correction for unequal variances. Differences are considered significant for *P* < 0.05. The dashed line represents the detection limit of the assay. Note that the lower viral load in the “no AT” group of infection with mCMV-Δm06 does not reflect virus attenuation but results from a lower initial dose of infection. Reproduced from [32] in a new arrangement. (**B**) Representative images of liver tissue sections stained by 2-color immunohistochemistry and hematoxylin counter-staining. (Red) Infected liver cells, mostly hepatocytes, identified by a cytoplasmic expression of m152 that is shared by all three viruses; (Black) Tissue-infiltrating CD8^+^ T cells. Bar marker: 50 μm. Reproduced from reference [32] in a new marker combination.

**Figure 5 viruses-14-00128-f005:**
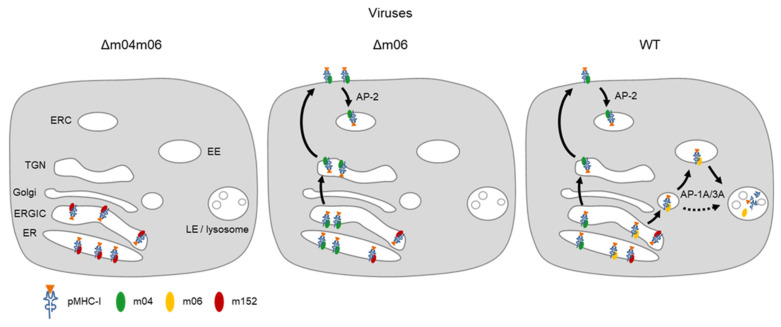
Graphical abstracts explaining agonistic and antagonistic vRAP functions. (ER), endoplasmic reticulum; (ERGIC), ER Golgi-intermediate compartment; (TGN), trans-Golgi network; (ERC), endosomal recycling compartment. Note that an early gene function of mCMV blocks cargo recycling from the ERC to the cell surface [103]; (EE), early endosome; (LE), Late endosome; AP, cellular adapter protein. Solid arrows indicate main pathways; the dashed arrow indicates an alternative route.

**Table 1 viruses-14-00128-t001:** Members of the *m02* gene family. Experimentally determined molecular masses refer to the most prevalent glycosylated isoforms. (‒) not studied or not applicable; (Red) cytoplasmic tail YXXΦ (where Φ is a hydrophobic amino acid residue) cargo sorting motif for binding to cellular adapter proteins AP-2 and/or AP-4; (Blue) cytoplasmic tail di-leucine cargo sorting motif [D/E](X)XXXL[L/I] for binding to cellular adapter proteins AP-1 and/or AP-3; (NK) natural killer; (PDB ID) Protein Data Bank identification code; (TCR) T cell receptor; (TMD) transmembrane domain.

*Gene*/Protein	Molecular Mass	Protein Structure	Sorting Motif	Binding Partner (Cargo)	Receptor of Cell Surface Ligand	Role in Immunity	Ref.
*m02*	‒	‒	YRDL	‒	‒	none found	[22]
*m03;m03.5*	‒	‒	none	‒	not identified	‒	[23]
*m04*/gp34	34 kDa	PDB ID 4PN6 [24]	YRRF	MHC-I	Ly49-family NK cell receptors Ly49A, C, G2	NK cell evasion	[25,26,27,28,29,30]
					TCR	Presentation of pMHC-I	[31,32]
*m05*	‒	‒	YICL	‒	‒	‒	
*m06*/gp48	48 kDa	‒	EPLARLL	MHC-I	‒	CD8^+^ T cell evasion	[33,34,35,36]
*m07*	‒	‒	YGFF	‒	‒	‒	
*m08*	‒	‒	YGFL	‒	‒	‒	
*m09*	‒	‒	YGFL	‒	‒	‒	
*m10*	‒	‒	YGFL	‒	‒	‒	
*m11*	‒	‒	none	‒	‒	‒	
*m12*	40 kDa	PDB ID 5TZN	YRRRGF	NKR-P1B and isoforms	Clr-b or unknown	NK cell evasion	[37]
*m13*	‒	‒	none	‒	‒	‒	
*m14*	‒	‒	none	‒	‒	‒	
*m15* locus	‒	‒	none	‒	‒	NK cell evasion	[38]
*m16*	‒	‒	YAIL	‒	‒	‒	

**Table 2 viruses-14-00128-t002:** Members of the *m145* gene family. Experimentally determined molecular masses refer to prevalent glycosylated isoforms; (‒) not studied or not applicable.

*Gene*/Protein	Molecular Mass	Protein Structure	Binding Partner	Receptor of Cell Surface Ligand	Role in Immunity	Ref.
*m145*	53 kDa	‒	MULT-1	NKG2D	NK cell evasion	[56]
*m146*	‒	‒	‒	‒	‒	
*m150*	‒	‒	‒	‒	‒	
*m151*	‒	‒	‒	‒	‒	
*m152*/gp40	48 kDa/ 40 kDa	PDB ID 4G59 [57]	MHC-I	TCR	T cell evasion	[19,58,59,60]
			RAE-1	NKG2D	NK cell evasion;	[61,62,63,64]
			STING	‒	Innate immunity	[65]
*m153*	80 kDa	PDB ID 2O5N [66]	Clr-b	NKR-P1B	NK cell evasion	[67,68]
*m154*	60 kDa	‒	AP-1 cargo: CD18, CD47, CD48, CD54, CD84, CD155, CD162, CD229, CD262, CD270	CD244 (for CD48)	Broad immune evasion NK cells and CD8^+^ T cells	[69,70]
*m155*	60 kDa	‒	H60	NKG2D	NK cell evasion	[71,72]
			CD40	CD40L	CD4^+^ T cell evasion	[73]
*m157*	42-50 kDa	PDB ID 2NYK [74]; 4JO8 [75]		Ly49H	Activation of NK cells	[76,77,78,79,80,81,82]
*m158*	‒	‒	‒	‒	‒	

## Data Availability

The data presented in this study are available on request from the corresponding author.

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
