# Peer review of "Host-Adapted Gene Families Involved in Murine Cytomegalovirus Immune Evasion"

_viruses, 2022, doi:10.3390/v14010128_

Round 1

Reviewer 1 Report

In the review manuscript titled: Host-adapted Gene Families Involved in Murine Cytomegalovirus Immune Evasion, the authors review both the molecular evidence for the function of the MCMV proteins m04, m06 and m152 during infection as well as their potential roles in immune evasion in vivo. The review is comprehensive, well written and presents evidence to support the authors ongoing hypothesis regarding memory inflation in the context of immune evasion. I only have a few minor grammatical and textual suggestions.

Page 2 – As a consequence of adaptation to the respective mammalian host during the estimated …

Page 4 - The structural analysis included and revealed binding of m04Smith, m04G4, and m04W8211 to the MHC-I molecules Dd, Ld, and Dk [24]. This sentence is confusing. What structural analysis are the authors referring to?

Figure 4A – review would benefit from a discussion as to why there are lower viral loads with the dm06 mutant virus so as not to have to go back to the original paper to understand the significance of this mutant.

Page 10 - that are M45-Db CTL

Author Response

We thank the reviewer for the helpful comments and modified the manuscript accordingly. 

Reviewer 2 Report

The authors represented a nice review describing the role of m04, m06, and m152 in viral antigen presentation during acute and latent CMVs infection. Overall the manuscript is well written and informative.

I have only one minor suggestion that the authors should add a conclusion section to highlight a strong take-home message and provide some open questions that need to be addressed in future research.

Author Response

We thank the reviewer for the helpful comment. Therefore, we added a new paragraph "Conclusion & outlook" in the revised version.